# The Prevalence of Low Handgrip Strength and Its Predictors among Outpatient Older Adults in a Tertiary Care Setting: A Cross-Sectional Study

**DOI:** 10.3390/geriatrics7040074

**Published:** 2022-07-08

**Authors:** Manchumad Manjavong, Apichart So-ngern, Panita Limpawattana, Natapong Manomaiwong, Thanisorn Kamsuanjig, Chudapha Khammak, Pongsak Chokkhatiwat, Kamolthorn Srisuwannakit

**Affiliations:** 1Division of Geriatric Medicine, Department of Internal Medicine, Faculty of Medicine, Khon Kaen University, Khon Kaen 40002, Thailand; manchu@kku.ac.th (M.M.); natama@kku.ac.th (N.M.); k_thanisorn@kkumail.com (T.K.); fang_chuda@kkumail.com (C.K.); pongcho@kku.ac.th (P.C.); kamolsr@kku.ac.th (K.S.); 2Division of Sleep Medicine, Department of Internal Medicine, Faculty of Medicine, Khon Kaen University, Khon Kaen 40002, Thailand; apicso@kku.ac.th

**Keywords:** aging, muscle strength, polypharmacy, possible sarcopenia, tertiary care hospital

## Abstract

Background: Low muscle strength is linked to several adverse health outcomes. However, there are limited data regarding its prevalence and associated factors in Thai older adults. This study aimed to fill that gap. Methods: This cross-sectional study was conducted with patients aged ≥ 60 years at the outpatient clinic of the internal medicine department of a tertiary care hospital from April 2020 to December 2021. Patient characteristics were collected, and a handgrip dynamometer was used to measure handgrip strength (HGS). Low HGS was defined according to the 2019 recommendations of the Asian Working Group for Sarcopenia. Results: In total, 198 patients were recruited. The prevalence of low HGS was 51%. Median HGS was 17.8 kg and 27.7 kg in women and men, respectively. Every age per year increase, greater number of medications of any type, and lower Montreal Cognitive Assessment (MoCA) score were independent factors associated with low HGS, with adjusted odds ratios of 1.1, 1.2, and 0.9, respectively. Conclusions: Low HGS was prevalent among older patients in this setting, indicating a high degree of possible sarcopenia. As there were some modifiable factors associated with low HGS, routine measurement, medication review, and cognitive evaluation are recommended for early diagnosis and management.

## 1. Introduction

Muscle strength is defined as the ability of a muscle or muscle group to exert force against resistance [1]. Normally, a progressive loss of muscle strength occurs beginning at 40 years of age. This decline begins at a rate of about of 10–15% per decade and becomes more rapid at 70 years [2]. Low muscle strength is associated with physical frailty, disability, premature mortality, longer hospital stays, and other age-related health complications [3,4,5,6]. Measurement of handgrip strength (HGS) is an easy and noninvasive tool in clinical practice that can reflect overall muscle strength [4]. The reported prevalence of low muscle strength based on HGS has varied from 6–46% depending on setting, study design, and its definition [7,8,9,10,11]. For example, a cross-sectional household-based study among the oldest Brazilians reported a prevalence of 39.2% [1], whereas a cross-sectional study among rural community-dwelling older women (age ≥ 60 years) in Southern Brazil found a prevalence of only 18.8% [9]. Another study reported a 6% prevalence among community-dwelling Chinese older adults based on CHARLS data [8], while a longitudinal study in the same country reported a 6% prevalence at baseline and 18.57% prevalence at a four-year follow-up [11]. A study in central Thailand found that about one-third (36.62%) of subjects aged over 60 had low HGS [10].

Studies from various countries and settings have identified several factors associated with muscle strength in older adults including (1) demographic factors (age, gender, education level), (2) lifestyle factors including anthropometric measures (e.g., body mass index, waist circumference, mid-arm circumference), physical activity, alcohol consumption, and vegetarian diet, and (3) comorbid conditions (diabetes mellitus, arthritis, cognitive impairment, previous falls, and number of medications) [3,4,5,6,7,9,11,12]. In 2019, the Asian Working Group for Sarcopenia (AWGS) reported low muscle strength and low physical performance as possible indicators of sarcopenia and recommended using HGS to represent muscle strength. In addition, they revised some of the cutoffs used for the HGS [13]. According to the AWGS, there are limited data available regarding the prevalence of HGS and its related factors in outpatient clinics of tertiary care hospitals, where the patients are more likely to have complex diseases than in community settings. These data help improve public awareness regarding low muscle strength as an indicator of possible sarcopenia. In addition, modification of related factors might delay its progression. The objectives of this study were thus to determine the prevalence of low HGS and its correlated factors in the outpatient clinic of a tertiary care hospital.

## 2. Materials and Methods

### 2.1. Study Design and Population

This cross-sectional study was part of a project entitled “Prevalence of frailty and related adverse events in older patients of Internal Medicine outpatient clinic”. The study population included patients aged 60 years or over (the Thai definition of older adults) who attended the outpatient clinic of the Srinagarind Hospital Internal Medicine Department (Khon Kaen University, Khon Kaen, Thailand), from January 2019 to December 2021. The study excluded patients with known illnesses that could inhibit test performance, who could not stand upright, with severe aphasia, with dementia, or who could not speak Thai.

### 2.2. Muscle Strength

A handgrip dynamometer (GRIP-D T.K.K.5401) was used to measure HGS. Patients sat upright with full elbow flexion. They were asked to use the dominant hand to hold the hand dynamometer tightly and squeeze as strongly as they could. Their best performance from three trials was used for our results. Low HGS was defined based on the AWGS 2019 guidelines as <28 kg for men and <18 kg for women [13].

### 2.3. Tools

#### 2.3.1. Patient Health Questionnaire (PHQ)-9

The PHQ-9 is a self-administered questionnaire to screen and monitor the severity of depression and response to treatment. It consists of nine items linked to the DSM-IV diagnostic criteria for major depressive disorder. The Thai version of the PHQ-9 has satisfactory internal consistency (Cronbach’s alpha 0.79) and moderately convergent validity to the Hamilton Depression Rating Scale (HAM-D; r = 0.56; *p* < 0.001). At a cut point of ≥9, it was found to have an area under the curve (AUC) of 0.89, sensitivity of 0.84, and specificity of 0.77. The positive predictive value (PPV), negative predictive value (NPV), and positive likelihood ratio were 0.21, 0.99, and 3.71, respectively [14].

#### 2.3.2. Montreal Cognitive Assessment (MoCA)

The MoCA is a 30-question screening test designed to detect cognitive impairment. It measures several cognitive domains including visuospatial executive function, naming, memory, attention, language, abstraction, delayed recall, and orientation. The total possible score is 30, with a higher score indicating worse cognitive performance. The Thai version of the MoCA has excellent internal consistency (Cronbach’s alpha coefficient 0.914), and the optimal cut-off point for detecting mild cognitive impairment (25/30) has a sensitivity of 80% and a specificity of 80% [15].

#### 2.3.3. Pittsburgh Sleep Quality Index (PSQI)

The PSQI is a self-report questionnaire that assesses sleep quality over the past month. An overall score > 5 indicates poor sleep quality (sensitivity 77.78% and specificity 99.33%). The Thai version of the PSQI has good internal consistency (Cronbach’s alpha coefficient 0.837), and the intraclass correlation coefficient was found to be 0.89 in all patients and 0.84 in patients with sleep disorders [16].

### 2.4. Procedure

Participants were selected using convenience sampling. After informed consent was obtained, a team of researchers collected data using interviewer-administered questionnaires and medical record review. Data collected included age, sex, educational level, marital status, body mass index, comorbid illnesses, depressive symptoms (defined by PHQ-9 ≥ 9), cognitive function according to the MoCA, number of medications, history of hospital admission in the past 12 months, history of fall in the past six months, alcohol and smoking status, and sleep quality (PSQI > 5 was considered to indicate poor sleep quality). HGS was measured using a handgrip dynamometer as described above according to AWGS 2019 recommendations [13].

### 2.5. Sample Size Calculation

Sample size calculations were based on the estimated prevalence of low HGS in older patients. The population proportion was calculated using a specified absolute precision formula. A previous study found the prevalence of low HGS in Thailand to be 36.62% [10], thus necessitating a sample size of 90 participants to obtain an alpha (α) of 0.05, Z (0.975) of 1.96, and error of 0.1. A total of 198 cases were included based on the objective of the main project.

### 2.6. Statistical Analysis

Descriptive statistics were used for demographic variables, which were expressed as median, interquartile range, and prevalence. Factors associated with low HGS were assessed using univariate and multivariate regressions analysis. Variables with statistical significance of at least 20% (*p*-value < 0.20) according to univariate analysis or that were found to be significant in previous studies were included in the multiple logistic regression analyses. Significance was set at 5% (*p*-value < 0.05). Adjusted odds ratio (AOR) and 95% CI were used to determine the strength of association. Data analysis was performed using R version 4.1.2.

### 2.7. Ethical Considerations

All experimental procedures were conducted in accordance with the Declaration of Helsinki. The study was approved by the Khon Kaen University Faculty of Medicine’s ethics committee (HE631065).

## 3. Results

Of the 198 patients enrolled, 51% had low HGS (51.6% in women and 50% in men). Median HGS was 17.8 kg (IQR 14.8–20) in women and 27.78 kg (IQR 22.4–31.8) in men. The characteristics of the studied population are shown in Table 1. Most of the patients had fewer than 12 years of education. Those with low HGS had lower MoCA and higher hospitalization over the past 12 months.

### Factors Associated with Low HGS

Associations between low HGS and demographic data were evaluated using univariate and multivariate analysis (Table 2). Age, education level, underlying diseases involving cerebrovascular accident, COPD/asthma, MoCA, number of medications, admission over the past 12 months, and current alcohol consumption all had *p*-values < 0.2 and were thus included in the multivariate analysis. Factors associated with low HGS in the final model were age (AOD 1.1), MoCA score (AOD 0.9), and number of medications (AOD 1.3).

## 4. Discussion

This is the first study to assess HGS and its related factors among older outpatients in a tertiary care hospital in Thailand. The results revealed a high prevalence of low HGS in this population (51%), implying a considerable proportion of older adults with possible sarcopenia. The prevalence of low HGS in this study was much higher than in previous reports (6–46%, as mentioned above) [7,8,9,10,11]. Median HGS was 17.8 kg in women and 27.7 kg in men, which was in line with a prior report of community-dwelling older adults in central Thailand (16 kg in women and 24 kg in men) [12] and in Taiwan (15.5 kg in women and 25.4 kg in men) [6]. However, our findings contrast with those of studies in some other Asian countries. A longitudinal study among community-dwelling Chinese older adults, for example, found female and male HGS to be 23.97 and 36.93 kg at baseline and 22.54 and 31.93 kg at a four-year follow-up [11]. Another study of older adults in a community setting in Sri Lanka found mean HGS to be 10.23 kg in women and 15.84 kg in men [4]. Differences in body build, ethnic background, lifestyle, socio-environmental factors, and study design (including the definition of low HGS) may have contributed to these differences. However, the fact that men had greater HGS than women in our study was consistent with previous findings [3,4,6,7,9,10,11].

This study found that older age and greater number of medications increased the risk of low HGS, whereas a higher MoCA score (better cognitive performance) was a protective factor. Older age has been identified as an independent factor associated with low HGS in many studies [3,6,8,9,12]. Age-related changes in muscle strength are the result of reductions in the potential activation of motor units and agonist muscles, and increases in the coactivation of antagonist muscles. Additionally, there are changes in capillary density and muscle architecture, causing a decrease in the total number of fibers. Fast-twitch fibers also appear to decline faster than slow-twitch fibers, and there is a decrease in the cross-sectional area of the muscle fibers [3,9]. Furthermore, age-related loss in muscle strength is usually associated with a multifactorial process consisting of reduced physical activity, malnutrition, and chronic illness [3]. Hence, strategies focusing on the prevention of muscle strength deterioration in this population should be considered. Taking a large number of medications was another factor associated with low HGS. This finding was consistent with those of a cross-sectional study among a community-dwelling older adults in Spain, and with a systematic review examining the association between polypharmacy and physical function in older adults [3,17]. The median number of medications in patients with low HGS in our study was five (i.e., polypharmacy [17]), compared to four in those without. Previous studies have found polypharmacy to be associated with physical function including muscle strength [17,18,19]. This might be explained by drug-drug and drug-disease interaction, side effects, and/or inappropriate dosages of medications such as sulfonylureas, glinides, statins, corticosteroids, and chemotherapy medications [17,18]. Further research should investigate in greater detail the possibility of medication causing low muscle strength. In addition, the pharmacodynamics of individual medications might act as mediators of the unfavorable effects of polypharmacy on muscle strength. For example, particular classes of medications depress the central nervous system, causing a reduction in physical and mental function. Conversely, muscle strength, which is a component representing physical function, could affect the number of medications administered to a patient. This is because low physical function is a risk factor for numerous chronic illnesses, affecting individual clinical conditions, the risk of chronic diseases, and multimorbidity [17,19]. Improving physical function may thus minimize the risk of polypharmacy and optimize drug prescription, leading to fewer adverse health outcomes in older patients.

Lower cognitive performance, according to the MoCA, was related to low HGS in this study. This finding supports those of earlier observational and interventional studies [7,9,20,21]. A probable explanation is that muscle strength might signify the integrity of nervous system activity [7,9,21]. One study among patients aged 60 years or over in the US showed that muscle strength was related to motor speed and visual-spatial processing, and another in the Netherlands in patients 85 (35% men) and 89 (29% men) years of age demonstrated an association between HGS and processing speed and memory [20]. However, as MoCA scores represent overall cognitive performance, our study was not able to examine associations with individual cognitive domains. High inflammatory markers, oxidative stress, and low levels of anabolic hormones (such as sex steroids and myokines) could also lead to lower muscle strength and cognitive function. Furthermore, older adults with cognitive dysfunction tend to have a sedentary lifestyle and insufficient dietary intake, which contribute to declines in muscle mass and strength [7,9,21].

Since life expectancy in general is increasing, the number of individuals with disabilities who survive into old age could be expected to increase too, thus contributing to the overall increase in the elderly population with disabilities. Muscle strength plays an essential role in the effective functioning of older adults since it is related to the integrity of the nervous system and the reverse is associated with adverse health consequences [3,4,5,6,7,9,21]. This study highlights the high prevalence of low muscle strength in older adults, and modifiable factors associated with low handgrip strength were identified. Psychomotor performance is one of these factors which is linked to several treatable causes such as depression, infection, medication, and metabolic abnormalities. To further increase the life expectancy of people in this age group while also maintaining a good quality of life, the measurement of handgrip strength, cognitive assessment, and medication reconciliation should be routinely evaluated in an outpatient setting.

There were some limitations to this study. First, because of the nature of the study design, the temporal relationship between outcome and exposure cannot be determined. Second, there were some factors previously found to be related to low HGS that were not significant in this study. This could be due to the relatively small sample size and differences in study design. Third, the association between cognitive function measured by the MoCA was only marginally significant (*p*-value = 0.04) and might thus have been due to chance. Additional cognitive tests conducted in the same sample may confirm the association. In addition, comorbidities related to osteoarticular pathology that may bias the outcome of the tests, such as osteoarthritis in the hand or dominant arm, were not considered. Finally, some factors that might be related to low HGS were not examined in this study, such as type of drug prescribed, or physical exercise performed. Further study is thus required to confirm our findings.

## 5. Conclusions

Low HGS was found in about a half of older outpatients at our tertiary care hospital, indicating a high prevalence of possible sarcopenia. Every age per year increase, number of medications, and poorer cognitive performance were independent factors related to low HGS. We recommend routine measurement of HGS in this setting, as it is simple to administer and might provide useful data regarding patient health status. In addition, intervention focusing on modifiable factors, including medication review and cognitive assessment, may be valuable in diminishing the risk of adverse health outcomes.

## Figures and Tables

**Table 1 geriatrics-07-00074-t001:** Demographic data of the studied population.

Variables	N = 198	
	Low HGS N = 101	Normal HGS N = 97
Age (year), med (IQR)	70 (67, 74)	66 (62, 69)
Sex, n (%)		
- Female	64 (51.61)	60 (48.39)
- Male	37 (50)	37 (50)
BMI (kg/m^2^), n (%)		
- Obese	41 (40.6)	40 (41.2)
- Overweight	21 (20.8)	25 (25.8)
- Normal	31 (30.7)	27 (27.8)
- Underweight	8 (7.9)	5 (5.2)
Years of education, n (%)		
- <12 years	81 (80.2)	61 (62.9)
- ≥12	20 (19.8)	36 (37.1)
Marital status, n (%)		
- Married	62 (61.4)	67 (69.1)
- Other *	39 (38.6)	30 (30.9)
Underlying diseases, n (%)		
- HT	62 (61.4)	66 (68.0)
- DM	37 (36.6)	36 (37.1)
- CVA	7 (6.9)	9 (9.3)
- COPD/asthma	9 (8.9)	3 (3.1)
- CKD	17 (16.8)	17 (17.5)
Depressive symptoms, n (%)	3 (2.97)	1 (1.0)
MoCA, med (IQR)	18 (14, 22)	22 (17, 24)
No. of medications used, med (IQR)	5 (1, 7)	4 (2, 5)
Admission within 12 months, n (%)	39 (38.6)	30 (30.9)
Falls within 6 months	15 (14.9)	13 (13.4)
Current alcohol consumption, n (%)	8 (7.9)	18 (18.6)
Current/ex-smoker, n (%)	29 (28.7)	27 (27.8)
Poor sleep quality, n (%)	42 (41.6)	39 (40.2)

Note: n: total number of participants, HGS: handgrip strength, med: median, IQR: interquartile ranges, BMI: body mass index (obese 25 kg/m^2^, overweight 23–24.9 kg/m^2^, normal 18.5–22.9 kg/m^2^, underweight < 18.5 kg/m^2^), HT: hypertension, DM: diabetes mellitus, CVA: cerebrovascular accident, COPD: chronic obstructive pulmonary disease, CKD: chronic kidney disease, MoCA: Montreal Cognitive Assessment, No.: number. * Other marital statuses include single, divorced, and widowed. Poor sleep quality was defined as Pittsburgh Sleep Quality Questionnaire (PSQI) score > 5. Depressive symptoms were defined as Patient Health Questionnaire (PHQ)-9 score ≥ 9.

**Table 2 geriatrics-07-00074-t002:** Factors associated with low HGS according to univariate and multivariate analysis.

Variables	Univariate			Multivariate		
	Crude OR	(95% CI)	*p*-Value	Adjusted OR	(95% CI)	*p*-Value
Age (years)	1.1	1.1–1.2	<0.001	1.1	1.06–1.2	<0.001 *
Sex						
- Female	1	-	-			
- Male	0.9	0.5–1.7	0.8			
BMI (kg/m^2^)						
- Obese	0.9	0.5–1.8	0.7			
- Overweight	0.7	0.3–1.6	0.4			
- Normal	1	-	-			
- Underweight	1.4	0.4–5.1	0.6			
Education ≥ 12 years	0.4	0.2–0.8	0.01	1.01	0.4–2.5	0.9
Marital status						
- Married	1	-	-			
- Other	1.4	0.8–2.5	0.3			
Underlying diseases						
- HT	0.7	0.4–1.3	0.3			
- DM	0.9	0.6–1.8	0.9			
- CVA	0.7	0.3–2.0	0.6	0.6	0.2–1.9	0.37
- COPD/asthma	3.1	0.9–14.1	0.1	1.6	0.4–8.4	0.5
- CKD	0.9	0.5–2.0	0.9			
Depressive symptoms	2.9	0.4–59.9	0.4			
MoCA	0.9	0.8–0.9	<0.001	0.9	0.8–0.9	0.04 *
No. of medication used	1.2	1.1–1.4	<0.001	1.3	1.1–1.4	<0.001 *
Admission within 12 months	1.4	0.8–2.5	0.3	1.4	0.7–2.7	0.4
Falls within 6 months	1.1	0.5–2.5	0.8			
Current alcohol consumption	0.4	0.1–0.9	0.03	0.7	0.2–1.9	0.4
Current and ex-smoker	1.0	0.6–1.9	0.9			
Poor sleep quality	1.1	0.6–1.9	0.8			

Note: OR: odds ratio, CI: confidence interval, BMI: body mass index (obese 25 kg/m^2^, overweight 23–24.9 kg/m^2^, normal 18.5–22.9 kg/m^2^, underweight < 18.5 kg/m^2^), HT: hypertension, DM: diabetes mellitus, CVA: cerebrovascular accident, COPD: chronic obstructive pulmonary disease, CKD: chronic kidney disease, (PHQ)-9 ≥ 9, MoCA: Montreal Cognitive Assessment. * Other marital statuses include single, divorced, and widowed. Poor sleep quality was defined as Pittsburgh Sleep Quality Questionnaire (PSQI) score > 5. Depressive symptoms were defined as Patient Health Questionnaire (PHQ)-9 score ≥ 9.

## Data Availability

No new data were created or analyzed in this study. Data sharing is not applicable to this article.

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
