# Peer review of "The Prevalence of Low Handgrip Strength and Its Predictors among Outpatient Older Adults in a Tertiary Care Setting: A Cross-Sectional Study"

_geriatrics, 2022, doi:10.3390/geriatrics7040074_

Round 1

Reviewer 1 Report

The peer-reviewed article «The association of cognitive performance and muscle strength among outgoing older adults of a tertial care setting; a cross-sectional study» is devoted to an urgent topic and presents an urgent problem. It should be noted that the description of the study is structured and logical, as well as clearly substantiated research results.

Author Response

Thank you for your comment. The revision has been edited the language presentation by a native speaker as your recommendation.

Reviewer 2 Report

This is a cross-sectional study with a sufficient sample size. The min finding is a high percentage of individuals with low HGS.

Low HGS is associated with the number of medication prescriptions. Such an association is a surrogate and indicative of multi-morbidity. 

However, if several testings are conducted in the same sample, the level of significance must be adapted according to Bonferroni. Therefore, the association between cognitive function measured by means of the MoCA test is only 0.04 and might be a finding of chance.

This issue must be discussed in more detail. In Addition, the headline of the paper should be changed accordingly. For example: prevelance of low HGS in a sample of....

The association between cognitive function and HGS is too weak and should not be mentioned in a headline.    

Author Response

English language and style are fine/minor spell check required

Reply: Thank you for your comment. The revision has been edited the language presentation by a native speaker as your recommendation.

This is a cross-sectional study with a sufficient sample size. The min finding is a high percentage of individuals with low HGS.

Low HGS is associated with the number of medication prescriptions. Such an association is a surrogate and indicative of multi-morbidity. 

However, if several testings are conducted in the same sample, the level of significance must be adapted according to Bonferroni. Therefore, the association between cognitive function measured by means of the MoCA test is only 0.04 and might be a finding of chance. This issue must be discussed in more detail.

Reply: Thank you for your suggestion. I have added the issue in the discussion part (limitation section) as “Third, the association between cognitive function measured by the MoCA was only marginally significant (p-value = 0.04) and might thus have been due to chance. Additional cognitive tests conducted in the same sample may confirm the association”.

In Addition, the headline of the paper should be changed accordingly. For example: prevelance of low HGS in a sample of.... The association between cognitive function and HGS is too weak and should not be mentioned in a headline.    

Reply: I have changed the headline of the paper as you recommended as “The prevalence of low handgrip strength and its predictors among outpatient older adults in a tertiary care setting; a cross-sectional study”.

Reviewer 3 Report

Geriatrics 1752082

Thank you for the opportunity to review this manuscript. Loss of strength is one of the criteria most associated with vulnerability in older people, related to increased comorbidity, presence of geriatric syndromes, dependency and mortality. It is an interesting article despite the limitations of the study design. I feel that the authors could value the following suggestions:

Abstract:

Line 21: add space  between  strength(HGS)

Line 25: please

 “Advancing age, an increasing number of medications, and Montreal 24 Cognitive Assessment (MoCA) score”

The authors should specify what is old age, what is or how many are the number of drugs and the MoCA score.

Line 26: the manuscript is not related to sarcopenia

Introduction:

I feel that in the introduction it is necessary to expand on the justification of the relationship between the loss of strength and the affectation of the cognitive and emotional dimension.

Methods,

Line 76-78: How these variables were recruited?

Line 84: Did the subjects perform the test only once or several times? Which procedure was followed in the case of multiple performance, the average or the highest of the three?

Falls: How is this variable collected, is it self-reported by the subject?

Although the authors indicate that the sample is a convenience sample, could they calculate the sample size to see if there is enough?

The authors have not explained how they have proceeded with the analysis of variables such as advanced age (what is it?) or ranges in BMI for obesity and overweight, etc...

The authors should change the term Depression to Depressive symptoms as the screening scale does not diagnose a medical pathology.

Discussion,

Line 167: The authors do not analyse sarcopenia, they are only detailing grip strength, please, the authors should try to eliminate possible relationships that they have not analysed.

Line 182, the authors have not discussed anything related to depressive symptoms, this being the aim of the study and the title of the manuscript.

Line 185, loss of strength due to the ageing process is one thing, pathological loss is another. Please review this concept

Line 221: the authors have not analysed anything related to the level of physical activity, so could they adjust these statements?

I feel that the authors should add many more limitations to their study.

The first is that comorbidity related to osteoarticular pathology that may bias the outcome of the tests, such as osteoarthritis in the hand or dominant arm, has not been collected.

Furthermore, the authors have not collected the type of drugs prescribed and venture in the discussion to discuss the results according to pharmacological groups, which should be reviewed.

Nor have they collected data on the physical exercise performed, an aspect that should also be reviewed by the authors in the discussion section.

Conclusions,

Please remove the sentence related to sarcopenia

Please clarify “advancing age…

Author Response

Thank you for the opportunity to review this manuscript. Loss of strength is one of the criteria most associated with vulnerability in older people, related to increased comorbidity, presence of geriatric syndromes, dependency and mortality. It is an interesting article despite the limitations of the study design. I feel that the authors could value the following suggestions:

English language and style are fine/minor spell check required

Reply: Thank you for your comment. The revision has been edited the language presentation by a native speaker as your recommendation.

Abstract:

Line 21: add space between strength (HGS)

Reply: I have added space as your comment.

Line 25: please

 “Advancing age, an increasing number of medications, and Montreal Cognitive Assessment (MoCA) score” The authors should specify what is old age, what is or how many are the number of drugs and the MoCA score.

Reply: I have edited as “Every age per year increase, greater number of medications of any type, and lower Montreal Cognitive Assessment (MoCA) score were independent factors associated with low HGS, with adjusted odds ratios of 1.1, 1.2, and 0.9, respectively”.  

Line 26: the manuscript is not related to sarcopenia

Reply: According to the Asian Working Group for Sarcopenia (AWGS) 2019 defined either low muscle strength or low physical performance as possible sarcopenia and recommended using HGS, in general, to represent muscle strength, low HGS in this manuscript is then related to possible sarcopenia.

Introduction:

I feel that in the introduction it is necessary to expand on the justification of the relationship between the loss of strength and the affectation of the cognitive and emotional dimension.

Reply: Thank you for your comment. Since the purposes of this study were to determine the prevalence of low HGS and its correlated factors in an outpatient clinic of a tertiary care hospital. Therefore, the introduction mainly focused on the prevalence and associated factors in general. To make it consistent with the headline and the objectives of the study, I have changed the headline to “The prevalence of low handgrip strength and its predictors among outpatient older adults in a tertiary care setting; a cross-sectional study”.

Methods,

Line 76-78: How these variables were recruited?

Reply: The researchers collected the data by asking the patients to sit upright with full elbow flexion. The patients were asked to use the dominant hand to hold hand dynamometer tightly and squeeze as strongly as they could then use the best performance of three trials.

This information was written in Line 76-79.

Line 84: Did the subjects perform the test only once or several times? Which procedure was followed in the case of multiple performance, the average or the highest of the three?

Reply: The subjects performed only once since this is a cross-sectional study.

Falls: How is this variable collected, is it self-reported by the subject?

Reply: It is self-reported by the subject.

Although the authors indicate that the sample is a convenience sample, could they calculate the sample size to see if there is enough?

Reply: Sample size calculations were based on the objective of this study which was the estimated prevalence of low HGS in older patients. The population proportion was calculated using a specified absolute precision formula. A previous study found the prevalence of low HGS in Thailand to be 36.62% [10], thus necessitating a sample size of 90 participants to obtain an alpha (α) of 0.05, Z (0.975) of 1.96, and error of 0.1. A total of 198 cases were included based on the objective of the main project.

Therefore, the sample size was enough. I have added this section in the materials and methods part.

The authors have not explained how they have proceeded with the analysis of variables such as advanced age (what is it?) or ranges in BMI for obesity and overweight, etc...

Reply: For advancing age, I have changed the term to “every age per year increase”.

For the ranges of BMI, they were noted under table 1 and table 2 as “BMI; body mass index (obesity 25 kg/m2, overweight 23-24.9 kg/m2, normal 18.5-22.9 kg/m2, underweight <18.5 kg/m2)”.

The authors should change the term Depression to Depressive symptoms as the screening scale does not diagnose a medical pathology.

Reply: I have changed as your suggestion.

Discussion,

Line 167: The authors do not analyse sarcopenia, they are only detailing grip strength, please, the authors should try to eliminate possible relationships that they have not analysed.

Reply: According to the Asian Working Group for Sarcopenia (AWGS) 2019 defined either low muscle strength or low physical performance as possible sarcopenia and recommended using HGS, in general, to represent muscle strength, low HGS in this manuscript is then related to possible sarcopenia.

Line 182, the authors have not discussed anything related to depressive symptoms, this being the aim of the study and the title of the manuscript.

Reply: Because depressive symptoms were not significant in this analysis, but cognitive performance (MoCA score) was significant. Therefore, it was discussed in this part. In addition, as mentioned earlier that the purposes of this study were to determine the prevalence of low HGS and its correlated factors in an outpatient clinic in a tertiary care hospital. To make it consistent with the headline and the objectives of the study, I have changed the headline to “The prevalence of low handgrip strength and its predictors among outpatient older adults in a tertiary care setting; a cross-sectional study”.

Line 185, loss of strength due to the ageing process is one thing, pathological loss is another. Please review this concept

Reply: I have added more information as “Furthermore, age-related loss in muscle strength is usually associated with a multifactorial process consisting of reduced physical activity, malnutrition, and chronic illnesses”.

Line 221: the authors have not analysed anything related to the level of physical activity, so could they adjust these statements?

Reply: Since this study did not collect level of physical activity or current work status that might be related to low HGS, this issue was noted in the limitation part as “Finally, some factors that might be related to low HGS were not examined in this study, such as type of drug prescribed, or physical exercise performed. Further study is thus required to confirm our findings”.

I feel that the authors should add many more limitations to their study.

The first is that comorbidity related to osteoarticular pathology that may bias the outcome of the tests, such as osteoarthritis in the hand or dominant arm, has not been collected.

Furthermore, the authors have not collected the type of drugs prescribed and venture in the discussion to discuss the results according to pharmacological groups, which should be reviewed.

Nor have they collected data on the physical exercise performed, an aspect that should also be reviewed by the authors in the discussion section.

Reply: Thank you for the suggestion. I have added this information in the limitation part as “…In addition, comorbidities related to osteoarticular pathology that may bias the outcome of the tests, such as osteoarthritis in the hand or dominant arm, were not considered. Finally, some factors that might be related to low HGS were not examined in this study, such as type of drug prescribed, or physical exercise performed. Further study is thus required to confirm our findings.”

Conclusions,

Please remove the sentence related to sarcopenia

Reply: According to the Asian Working Group for Sarcopenia (AWGS) 2019 defined either low muscle strength or low physical performance as possible sarcopenia and recommended using HGS, in general, to represent muscle strength, low HGS in this manuscript is then related to possible sarcopenia.

Please clarify “advancing age…

Reply: I have changed it to “every age per year increase…”

Round 2

Reviewer 2 Report

all comments of the reviewer have been met. 

Author Response

Dear reviewer,

Thank you again for your suggestion.

Reviewer 3 Report

I feel that the authors have greatly improved the manuscript although I still have two points to suggest:

The first is that the discussion I feel that it could be better worked out. 

The second is in the methods section, in terms of obtaining the handgrip, I was referring to what study protocol the authors have used, not whether it is a cross sectional or a repeated measures study.

There are study protocols that take three handgrip measurements and choose the highest, others that take three measurements and take the mean and I understand that in this case they have only obtained one measurement? 

Author Response

I feel that the authors have greatly improved the manuscript although I still have two points to suggest:

The first is that the discussion I feel that it could be better worked out. 

Reply: I have added how to apply the finding in the discussion part (line 233-243) as “Since life expectancy in general gains, individuals with disabilities who survive into old age could be expected to contribute to the overall growing in the elderly population with disabilities. Muscle strength plays an essential role in effective functioning in older adults since it is related to the integrity of the nervous system as described earlier and associated with adverse health consequences. This study highlights the high prevalence of low muscle strength in older adults, and modifiable factors associated with low handgrip strength were identified. Psychomotor performance is one of these factors which is linked to several treatable causes such as depression, infection, medication, and metabolic abnormalities. To add life to a year in this age group in achieving good quality of life, measurement of handgrip strength, cognitive assessment, and medication reconciliation should be routinely evaluated in an outpatient setting”.

The second is in the methods section, in terms of obtaining the handgrip, I was referring to what study protocol the authors have used, not whether it is a cross sectional or a repeated measures study.

There are study protocols that take three handgrip measurements and choose the highest, others that take three measurements and take the mean and I understand that in this case they have only obtained one measurement?

Reply: This is a cross-sectional study. The data were obtained from one measurement.